# Identifying enhancer properties associated with genetic risk for complex traits using regulome-wide association studies

**Alex M. Casella**[1,2], **Carlo Colantuoni**[3], **Seth A. Ament**[1,4]*

**1** Institute for Genome Sciences, University of Maryland School of Medicine, Baltimore, Maryland, United States of America, **2** Medical Scientist Training Program, UMSOM, Baltimore, Maryland, United States of America, **3** Departments of Neurology and Neuroscience, Johns Hopkins University School of Medicine, Baltimore, Maryland, United States of America, **4** Department of Psychiatry, University of Maryland School of Medicine, Baltimore, Maryland, United States of America

* sament@som.umaryland.edu

**Data Availability Statement:** All enhancer annotation files are available at http://data.nemoarchive.org/other/grant/sament/sament/

## Abstract

Genetic risk for complex traits is strongly enriched in non-coding genomic regions involved in gene regulation, especially enhancers. However, we lack adequate tools to connect the characteristics of these disruptions to genetic risk. Here, we propose RWAS (Regulome Wide Association Study), a new application of the MAGMA software package to identify the characteristics of enhancers that contribute to genetic risk for disease. RWAS involves three steps: (i) assign genotyped SNPs to cell type- or tissue-specific regulatory features (e.g., enhancers); (ii) test associations of each regulatory feature with a trait of interest for which genome-wide association study (GWAS) summary statistics are available; (iii) perform enhancer-set enrichment analyses to identify quantitative or categorical features of regulatory elements that are associated with the trait. These steps are implemented as a novel application of MAGMA, a tool originally developed for gene-based GWAS analyses. Applying RWAS to interrogate genetic risk for schizophrenia, we discovered a class of risk-associated AT-rich enhancers that are active in the developing brain and harbor binding sites for multiple transcription factors with neurodevelopmental functions. RWAS utilizes open-source software, and we provide a comprehensive collection of annotations for tissue-specific enhancer locations and features, including their evolutionary conservation, AT content, and co-localization with binding sites for hundreds of TFs. RWAS will enable researchers to characterize properties of regulatory elements associated with any trait of interest for which GWAS summary statistics are available.

## Author summary

Enhancers are regulatory regions that influence gene expression via the binding of transcription factors. Risk for many heritable diseases is enriched in regulatory regions, including enhancers. In this study, we introduce a novel application of the MAGMA software tool that enables testing for associations between enhancer attributes and risk, and

RWAS. Code can be found at https://github.com/
casalex/RWAS.

**Funding:** This work was supported by two grants
from the National Institute of Mental Health
(https://www.nimh.nih.gov/): F30MH120910 (PI:
AMC) and R24MH114815 (PI: Ronna Hertzano,
University of Maryland School of Medicine). SAA,
AMC and CC all received salary support on the
R24MH114815, an aim of which is to perform
integrated analyses of single-cell genomic data
related to brain development. The funders had no
role in study design, data collection and analysis,
decision to publish, or preparation of the
manuscript.

**Competing interests:** The authors have declared
that no competing interests exist.

we use this method to determine the enhancer characteristics that are associated with risk for schizophrenia. We found that enhancers associated with schizophrenia risk are both evolutionarily conserved and in physical contact with mutation-intolerant genes, many of which have neurodevelopmental functions. Risk-associated enhancers are also AT-rich and contain binding sites for neurodevelopmental transcription factors.

## Introduction

Non-coding genomic regions such as enhancers and promoters, as well as the transcriptional machinery that interacts with them, govern the gene regulatory programs underlying the proper development and function of the body's tissues and organs [1,2]. Genetic variation influencing many human traits is enriched in these gene regulatory regions [3–8]. In genome-wide association studies (GWAS) of diseases such as cardiovascular, autoimmune, and neuro-psychiatric disorders, more than 90 percent of SNPs in risk loci are non-coding variants [9]. Epigenomic studies over the past decade have mapped tissue- and cell type-specific gene regulatory elements in the non-coding genome, opening the door for large-scale exploration of their contribution to human disease. These studies have demonstrated that disease-associated genetic variation is concentrated in regulatory regions in a tissue- and cell type-specific manner. For example, rheumatoid arthritis (RA) and Crohn's disease risk are highly enriched in regions of accessible (active) chromatin from blood and immune cells, while type 2 diabetes risk is enriched in open chromatin from endocrine tissue [3]. Disease risk has also been connected to more specific regulatory elements, including enhancers, which are distal gene regulatory elements that activate and refine the cell type- and context-specific activity of many promoters [10].

These findings suggest that much of the genetic risk for complex traits acts through the disruption of regulatory regions unique to the tissues and cell types that are most relevant in each trait. However, there remain substantial gaps in our knowledge about the mechanisms by which variants in specific promoters and enhancers predispose to risk. This is in part because existing tools, while powerful, are not designed to evaluate the features of specific regulatory regions that are associated with disease risk. Methods such as H-MAGMA and ABC focus on predicting the target genes of distal enhancers and use these predictions to predict causal genes at GWAS risk loci [11,12]. Epigenomic fine-mapping tools such as PAINTOR and RiVIERA integrate non-coding annotations to predict specific, causal SNPs [13,14]. Stratified Linkage-Disequilibrium Score Regression (LDSC) performs genome-wide inference of genomic features (e.g., open chromatin regions, evolutionarily conserved regions) enriched for disease risk, but is primarily used to assess binary annotations–rather than quantitative scores–and is underpowered for annotations representing less than ~1% of the genome [3]. FENRIR tests for associations between disease risk and networks of enhancers with similar features [15].

Here, we propose RWAS (for Regulome-Wide Association Study) as an application of the MAGMA software suite to test associations of genetic risk with specific enhancers and enhancer properties. In the RWAS framework (Fig 1), we first collect enhancer annotations in a tissue relevant to the trait of interest, then identify specific risk-associated enhancers by aggregating the effects of all SNPs that overlap the enhancer's position in the genome. Finally, we test for associations of enhancer features with disease risk using a regression framework. RWAS is implemented as a novel application of MAGMA [9], which was originally developed for gene-based association studies and is widely used for that purpose. RWAS is computationally efficient and readily extensible to any trait for which GWAS summary statistics are

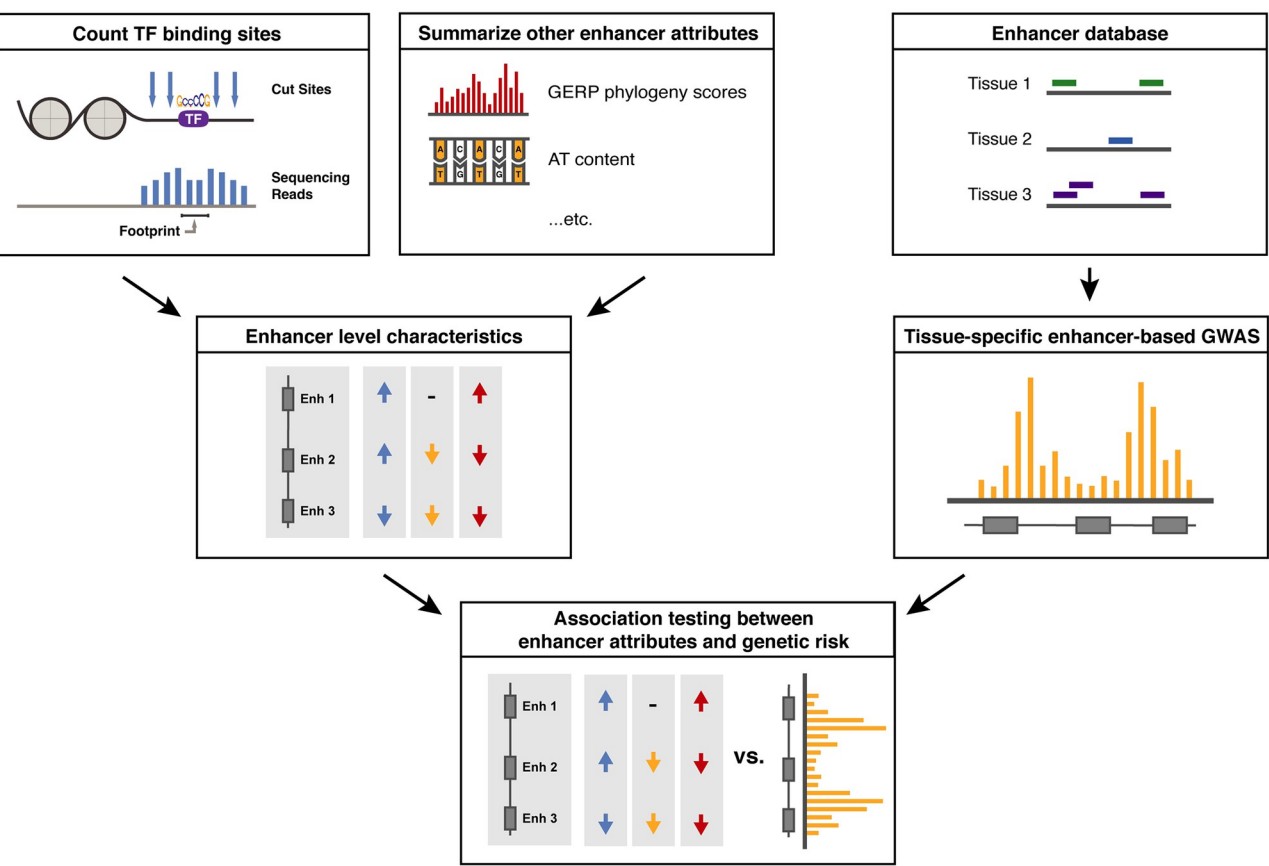

**Fig 1. RWAS workflow overview.** In brief, the RWAS workflow involves annotating SNPs to enhancers and other regulatory regions (rather than genes). Enhancer-level summary statistics are computed for input into association testing. Then, we use the MAGMA linear modeling framework to compute genetic associations between supplied enhancer-level covariates and these enhancer-based GWAS summary statistics. This approach relies on high-quality enhancer annotations for the tissue of interest that capture genetic risk for the disorder. To ensure these conditions were met, we first thoroughly characterized a set of brain-specific enhancers and demonstrated that these enhancers capture genetic risk for schizophrenia.

available. We apply RWAS to characterize enhancers and enhancer features that are associated with risk for schizophrenia (SCZ), a severe psychiatric disorder for which well-powered GWAS identified hundreds of risk loci enriched in brain-specific gene regulatory regions [5, 9]. As part of this work, we also compiled a resource of high-quality adult and fetal brain enhancer maps to identify risk-associated enhancer traits in the brain. Our analyses reveal novel associations of SCZ risk with AT-rich enhancers in the developing brain and risk-associated transcription factor networks.

## Results

### A database of enhancers and enhancer annotations in the human brain

The three elements required for an RWAS analysis are a database of tissue-specific gene regulatory elements, annotations describing the attributes of the enhancers for association testing, and GWAS summary statistics for a trait of interest. Here, as a regulatory element database, we utilized ChromHMM-derived enhancer predictions in 127 human tissues and cell types from the ROADMAP consortium [16]. Specifically, we utilized a 25-state model that integrated data from 12 histone marks and related genomic features [17]. Enhancers predicted by ChromHMM have been extensively validated in independent epigenomic datasets, and their

tissue-specific activity predicts the expression of nearby genes [17]. A major advantage of the ROADMAP dataset is that the wealth of different tissues available makes cross-tissue comparison easier, enabling an unbiased view of enhancer activity across tissues and cell types. Analyses presented in this paper focus primarily on schizophrenia (SCZ), for which purpose we are primarily interested in annotations of enhancers in the brain. The dataset contains chromatin state annotations for 15 brain-related samples, including seven samples from the adult brain, three from the prenatal brain at mid-gestation, three from embryonic stem cell (ESC)-derived neuronal progenitors or neurons, and two from neurosphere cultures.

We validated these enhancer annotations by four approaches. First, we tested for overlap of ChromHMM-predicted brain enhancers with enhancers predicted by ChIP-seq in independent samples. Consistent with previous analyses of ChromHMM-derived enhancers, we found that the enhancers utilized in our analysis were enriched for regions marked by acetylation at lysine 27 of the histone 3 tail (H3K27ac), which marks active regulatory regions, and depleted for tri-methylation at lysine 9 on the histone 3 tail (H3K9me3), which marks heterochromatin (S1 Fig).

Second, we compared enhancer locations in the 127 samples on the basis of summary statistics, including enhancer length, genomic coverage, enhancer number, and AT-richness (S2 Fig). Brain enhancers were largely similar to other tissues in terms of number, length, and coverage (S2A, S2C and S2E Fig). Within the brain, adult brain samples had the highest coverage and number of predicted enhancers, while samples from fetal brain and models of neurodevelopment had lower coverage and number of enhancers (S2B and S2D Fig). Fetal brain and neurosphere samples had average enhancer lengths nearly 50 bp longer than those in adult brain samples (S2F Fig).

Third, we tested whether these enhancer annotations capture an element of tissue specificity. The Jaccard index was used to quantify pairwise similarity among the genomic locations of enhancers utilized in the 127 samples. As expected, enhancer utilization clustered samples by organ, as well as by developmental age (S3 Fig). In the brain, we found three groups of samples distinguished by their enhancer utilization, corresponding to adult brain, fetal brain and cultured neurospheres, and cultured neural progenitors (Fig 2).

Fourth, we tested that our enhancer annotations confirm known associations, focusing on SCZ [5]. Previous studies have shown that enhancers and other gene regulatory regions active in the human brain are enriched for heritability in SCZ [3–5]. As expected, stratified LD Score Regression using summary statistics from schizophrenia GWAS [5] confirmed that brain enhancers from our analysis were highly enriched for SCZ risk (Fig 3A). The adult brain-specific enhancer annotation most significantly enriched for SCZ risk was the inferior temporal gyrus (sample E072, p = 4.6E-14), and the fetal brain-specific enhancer annotation most significantly enriched for risk was female fetal brain (sample E082, p = 1.28E-9). These enrichments were comparable in significance to the enrichment of SCZ risk in two sets of adult prefrontal cortex enhancers from the PsychENCODE consortium (Fig 3B) [18]. In summary, our validation tests indicate that ROADMAP ChromHMM models provide robust annotations of enhancers in the fetal and adult brain that capture a tissue-specific element of genetic risk for SCZ. These analyses define a total of 388,011 non-overlapping enhancer regions and are available at http://data.nemoarchive.org/other/grant/sament/sament/RWAS.

## RWAS reveals enhancers and enhancer characteristics associated with risk for schizophrenia

We hypothesized that SCZ risk is associated with SNPs that impact specific enhancers that are active in the brain. To identify these enhancers, we performed an "enhancer-based" GWAS

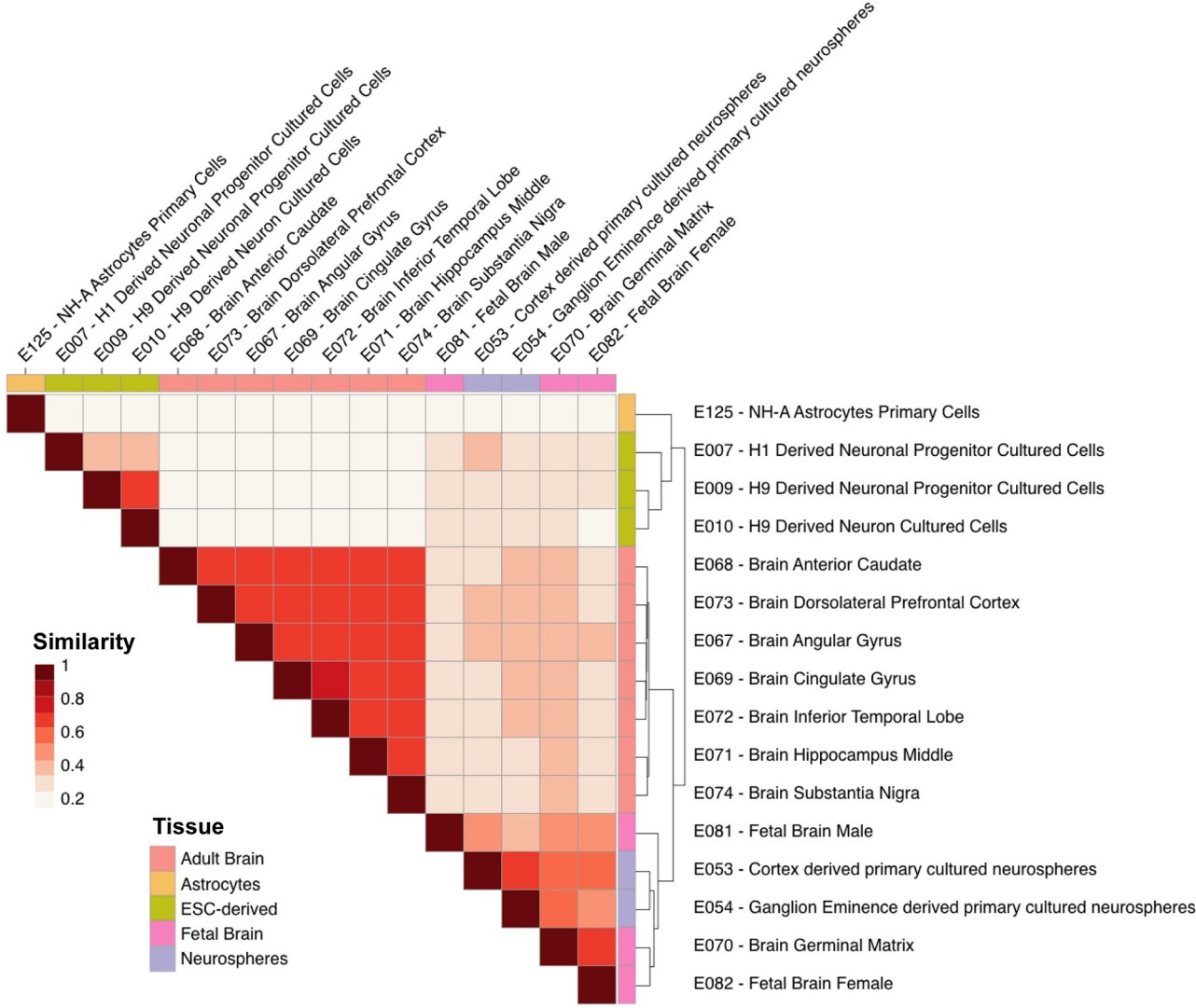

**Fig 2. Genome-level Jaccard similarity matrix demonstrates age- and experimental model- specific enhancer patterning.** Fetal brain and neurosphere samples cluster together when the tree is cut at the second level, while adult brain samples, ESC-derived clusters, and astrocytes form separate groups. Color denotes Jaccard similarity statistic. Groupings determined using hierarchical clustering.

analysis of the PGC2 SCZ GWAS, testing for significance of the aggregated SNPs within each enhancer using the SNP-wise regression model implemented in MAGMA. Fig 4A illustrates how enhancer-based GWAS annotates signal peaks from a SNP-based GWAS (top) to specific disease-associated enhancers (bottom). This analysis revealed a total of 2,784 risk-associated enhancers at a genome-wide significance threshold $p < 1.3E-7$, which corresponds to alpha $< 0.05$ after Bonferroni correction for 388,011 non-overlapping brain-activated enhancer regions in our database (Fig 4B). 2,001 of these risk-associated brain enhancers are located within 63 of the 108 risk loci identified in the original (SNP-based) analysis of these data, while the remaining enhancers are at loci that did not reach genome-wide significance in the primary analysis. Examination of specific loci indicated that risk-associated enhancers capture the genetic risk signal at many of the SNP-based risk loci in a tissue-specific manner (Fig 4B). Overall, we found associations of SCZ risk with substantially more brain enhancers than with enhancers from other cell types. However, we also find loci at which enhancers from

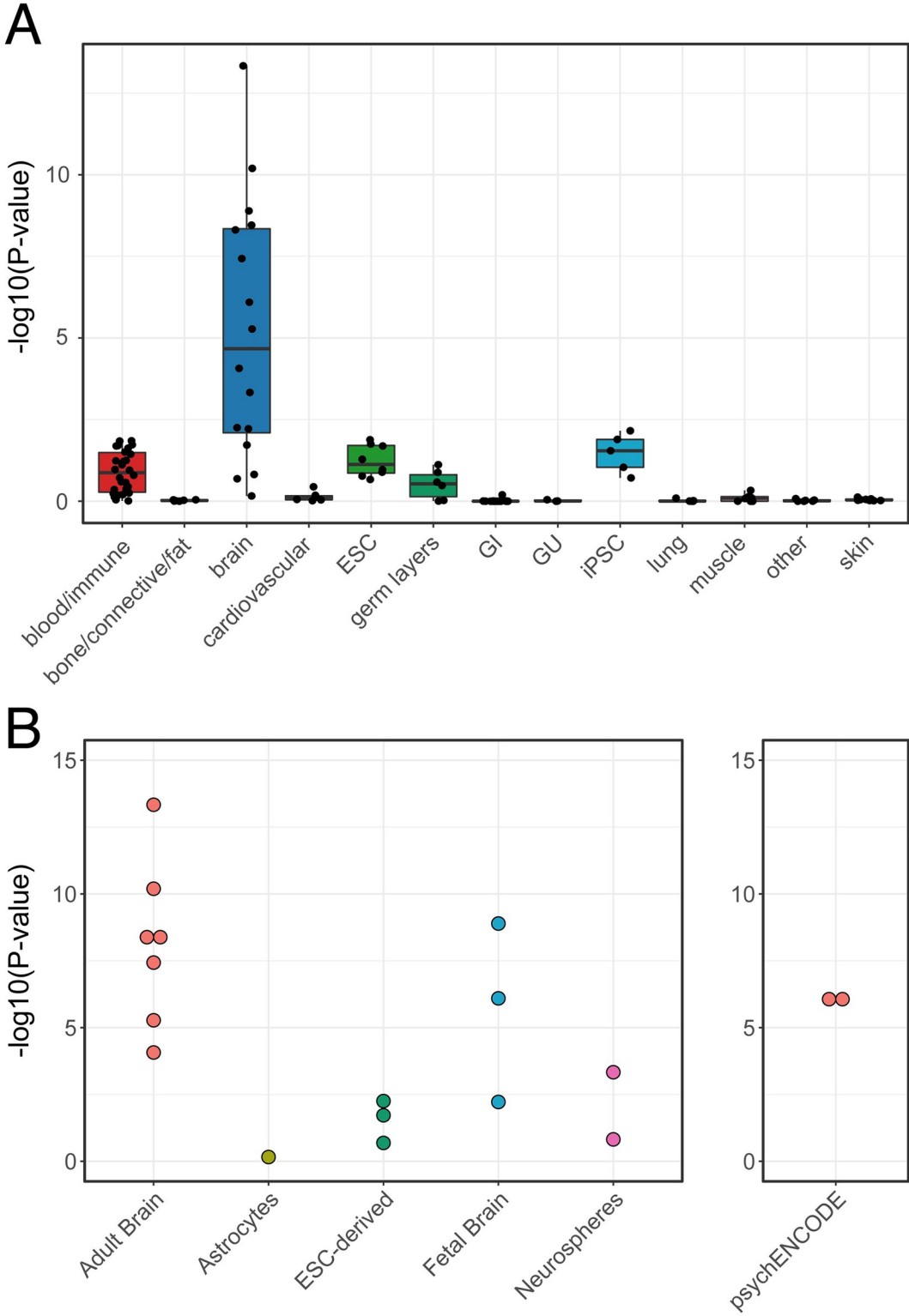

**Fig 3. Genetic risk for schizophrenia is enriched in ChromHMM-derived brain enhancers.** A) Partitioned heritability of enhancer annotations by tissue in schizophrenia. Brain enhancers are enriched for heritability in schizophrenia compared to other tissues. B) Partitioned heritability of individual brain samples. Adult brain enhancers had the most significant enrichment, followed by fetal brain enhancers.

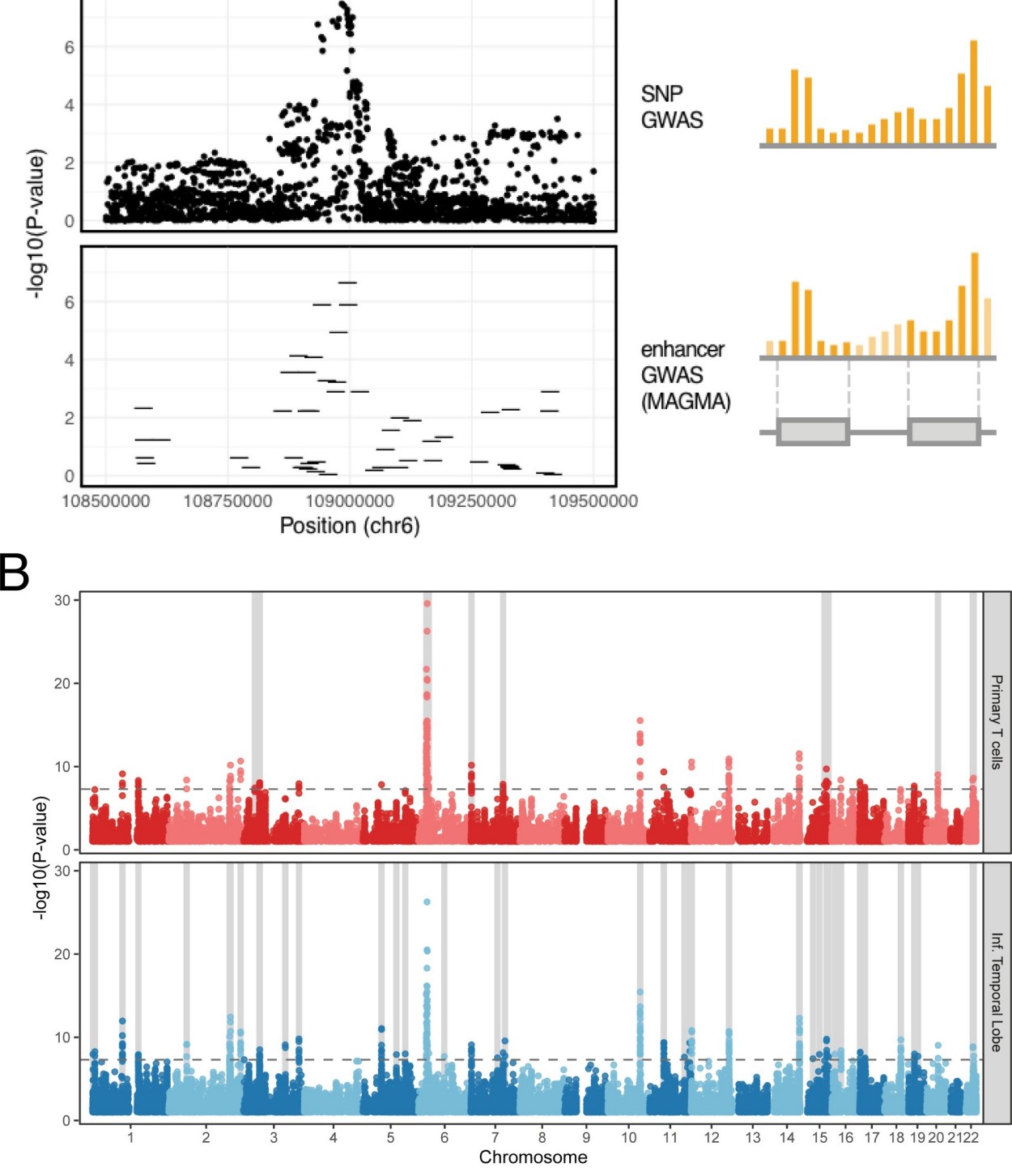

**Fig 4. Identification of brain-expressed enhancers associated with genetic risk for schizophrenia.** A) Enhancer-based GWAS allows the aggregation of non-coding SNPs into nearby enhancer regions and captures risk loci as enhancer-level risk associations. Individual points on the top panel denote SNPs, while the lines on the bottom panel represent enhancer regions. B) Brain enhancers capture risk loci missed by enhancers from other tissues. Enhancers from primary T cells captured fewer genome-wide significant signals when compared to enhancers from the inferior temporal lobe of the adult brain. The light gray shaded areas denote loci where a given enhancer annotation has more genome-wide significant enhancers when compared to the other annotation.

other tissue and cell types have the strongest p-values, potentially pointing to roles for these cell types in SCZ risk. For instance, we find associations with certain T-cell specific enhancers, potentially implicating immune cell types in SCZ risk (Fig 4B).

We compared the SCZ-associated enhancers from our analysis to SCZ-associated enhancers predicted by an alternative approach, FENRIR. We found that the ChromHMM enhancers used in our model had more significant enrichment for SCZ risk compared to the FENRIR networks. According to estimates from Chen et al. 2021, FENRIR brain enhancers had an LDSC enrichment significance of p = 2.7E-04, which is less significant than all but one of the 10 adult and fetal brain ChromHMM enrichments from our analyses [15]. While there was not a great deal of overlap between the two enhancer sets, enhancer effect sizes from our analysis correlated strongly with FENRIR scores. For example, 16,761 of the 105,489 male fetal brain enhancers identified in our analysis had a direct overlap with a FENRIR enhancer, and the effect size predicted by our model was highly correlated with FENRIR predicted disease association in schizophrenia (linear regression p<2E-16, beta = 0.035). The top 100 enhancers from our analysis with an overlapping FENRIR enhancer had FENRIR scores >4x higher than lower ranking enhancers (0.58 vs. 0.14). These results confirm that SCZ-associated enhancers identified in our analysis can be reproducibly associated with SCZ by an independent approach and suggest that our strategy may have greater statistical power.

To further validate SCZ-associated enhancers identified in our analysis, we tested for overlap with functionally-validated enhancers from massively parallel reporter assays (MPRA) of schizophrenia risk alleles [19] and with expression quantitative trait loci (eQTLs) in the prefrontal cortex [20]. For example, sixty-six of the SCZ-associated enhancers identified in our analysis of the fetal male brain contained SCZ-associated SNPs that were functionally validated to impact enhancer activity by MPRA. Permutation tests suggest that this overlap is substantially more than expected by chance (permutation p-values < 0.05 in all 10 brain samples). This analysis provided independent evidence for several of the top SCZ-associated enhancers in our analysis. A fetal brain-specific enhancer at chr1:243555100–243556100 (p = 7.68E-11), located in an intron of *SDCCAG8*, contains a SNP (rs77149735) associated with differential enhancer activation by MPRA. A second fetal brain-specific, risk-associated enhancer from our analysis, located at chr22:42657000–42658000 (p = 2.4E-9) contained the SNP rs134873, which was significantly associated with differential enhancer activity in MPRA assays and has been previously described as an eQTL for the genes *FAM109B*, *NAGA*, *LINC00634*, and *WBP2NL*. Similarly, we tested for overlap of SCZ-associated enhancers with cortex-specific eQTLs from the GTEx collection (v7) [21]. We found a strong positive correlation between eQTL status and SCZ risk across all 10 brain enhancer sets tested (p < 2e-07). These results demonstrate that many SCZ-associated enhancers identified in our analysis have strong evidence of regulatory impact in the brain.

Next, we tested the hypothesis that risk-associated enhancers regulate gene sets that have previously been implicated in neuropsychiatric studies. We used Hi-C data from the developing brain [22] to predict the targets of risk-associated enhancers. Across all the adult and fetal brain enhancer annotations, a total of 720 genes were in contact with at least one statistically significant risk-associated enhancer. These associations were quite reproducible: 648 of these genes were identified in more than one brain tissue enhancer annotation and 248 were found in all ten. Using these enhancer-gene maps, we tested for enrichments in 64 gene sets that have previously been implicated in SCZ risk (Fig 5A). Enhancer targets were enriched for genes that are intolerant of loss-of-function mutations (p = 2.37E-3, pLI; p = 2.16E-4, LOEUF [see Methods for definition]). Risk-associated enhancers also disproportionally contact genes that are bound by the neuron-specific RNA-binding proteins Fragile X mental retardation protein (FMRP) (p = 3.07E-5) and RBFOX1/3 (p = 3.75E-4), as well as targets of the autism-associated

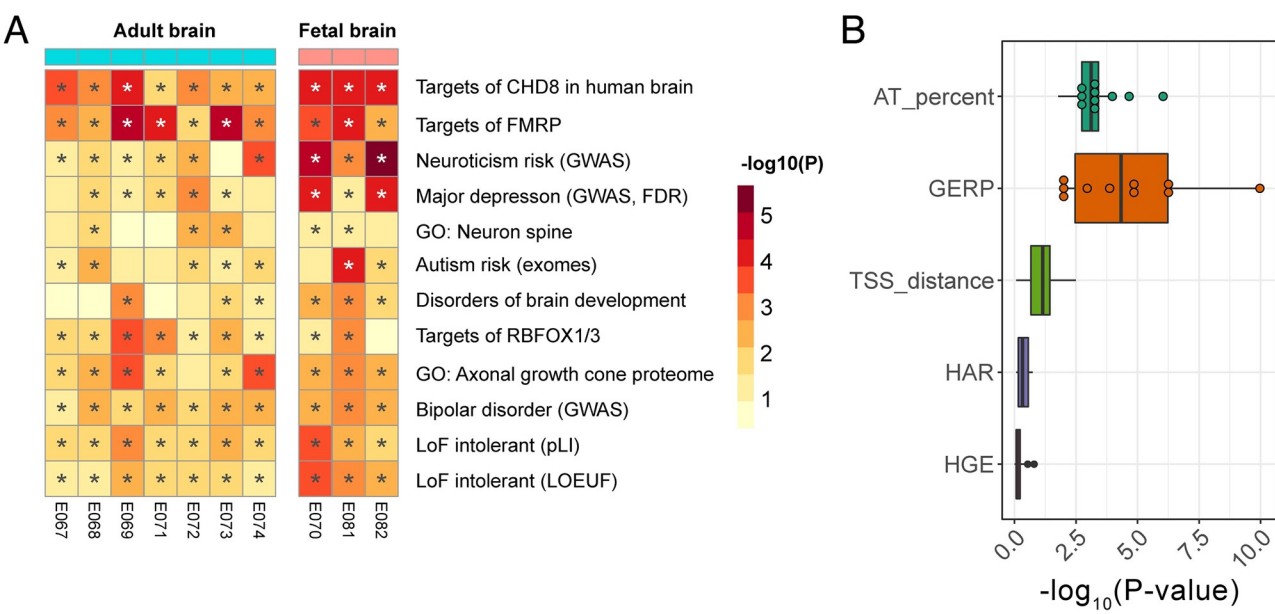

**Fig 5. Features of brain-expressed enhancers associated with schizophrenia risk.** A) Risk-associated enhancers are in physical contact with gene sets previously implicated in risk for neuropsychiatric disorders. Shown here are gene sets with a median p-value across the 10 brain enhancer samples < 0.05. GWAS = genes at GWAS risk loci. Full description of gene lists available in Methods. B) Evolutionary conservation and AT-richness of enhancers were associated with schizophrenia risk. HGE/HAR status and distance to the nearest gene TSS were not associated with risk.

chromatin remodeling gene Chromodomain-helicase-DNA-binding protein 8 (CHD8) (p = 3.99E-4). Rare mutations in FMRP and CHD8 cause neurodevelopmental disorders with autistic features [23–30] and regulate neurodevelopmental gene networks that have previously been linked to SCZ in genetic and proteomic studies [31,32]. These findings extend previous gene-based analyses [33,34].

Next, we tested the hypothesis that risk-associated enhancers differ in their evolutionary history from other enhancers in the brain. Enhancers with deep evolutionary conservation may have particularly important functions in the brain. It has also been postulated that risk for SCZ may involve evolutionarily novel enhancers, some of which regulate human-specific aspects of brain development [35,36]. Evolutionary conservation within enhancer regions (defined by GERP phylogeny scores) was positively associated with risk (Fig 5B), with fetal brain enhancers having the most significant associations (male fetal brain, 1.1E-10; germinal cortex at 20wk gestation, 5.2E-7; female fetal brain, 5.9E-7; all 10 adult and fetal brain enhancer annotations significant at FDR < 0.05). By contrast, two categories of evolutionarily novel enhancers–human accelerated regions (HARs) and human-gained enhancers (HGEs)–were not significantly associated with schizophrenia risk (Fig 5B), in agreement with previous results [37]. This finding is unlikely to be due to low power, since 12,501 brain enhancers were found within 5kb of an HGE and 7,984 were located within a HAR. Therefore, schizophrenia risk-associated enhancers are older in evolutionary time and are not generally under positive selection.

Since many enhancers regulate proximal promoter regions, we hypothesized that enhancers closer to a transcription start site would be more strongly associated with disease risk. However, we found that distance to the nearest gene was not associated with risk (Fig 5B). This is in line with the discovery of significant long-range interactions between schizophrenia risk SNPs and genes with neuronal functions [22].

Unexpectedly, one of the enhancer features most strongly associated with SCZ risk was the percent of adenine-thymidine base pairs (AT richness), which was positively associated with risk across all brain tissues surveyed (Fig 5B). The most significant association in adult brain was in prefrontal cortex enhancers (E073, p = 9.1E-7), while the strongest association in the fetal brain was in the fetal germinal matrix (E070, 20 weeks gestational age; p = 5.68E-6). Overall, brain enhancers do not have substantially higher AT richness than enhancers in other tissues (S4C Fig). In addition, we did not find a strong association between AT richness and SCZ risk within enhancers from other tissues (S5 Fig). Therefore, these results suggest that SCZ risk is enriched specifically at AT-rich enhancers in the adult and developing brain.

### SCZ-associated enhancers are enriched for binding sites for neurodevelopmental transcription factors recognizing AT-rich sequence motifs

We hypothesized that the association of AT rich enhancers with SCZ corresponds with occupancy by transcription factors that recognize AT-rich sequence motifs. To test this, we performed an RWAS testing for association between SCZ risk and binding sites for individual TFs. We used tissue-specific TF binding site predictions for 503 TFs, derived from integration of DNase-seq footprinting analysis in the human brain with JASPAR2016 vertebrate sequence motifs [38]. There was a strong association between the AT-richness of a given TF binding site motif and the effect size in our model (p = 2.0E-15 in female fetal brain). These associations were also borne out in a meta-analysis performed by combining all 10 adult and fetal brain enhancer RWAS (p < 2E-16). We also found that TF motifs with positive association with SCZ risk in our RWAS had higher AT percentages than motifs with a negative association; in other words, TF motifs that were overrepresented in risk-associated enhancers had higher AT percentages than TF motifs that were depleted in risk-associated enhancers (Fig 6A).

While the nucleotide composition of promoters and of larger chromosomal segments (isochores, >300 kb on average) has been extensively described, the functional differences between AT-rich vs. GC-rich enhancers are not well understood. Strikingly, many of the most positively associated sequence motifs, all of which are AT rich, are recognized by neurodevelopmental TFs, including members of the MEF2 family, the EMX family, and the DLX family (Table 1). Based on this result, we asked whether AT-richness might be a general feature of neurodevelopmental TFs. Indeed, TFs annotated to the Gene Ontology term "cell morphogenesis involved in neuron differentiation" and related GO terms had higher motif AT percentages than other TFs (Wilcoxon p = 5.0E-5, Fig 6B), and motifs recognized by these neurodevelopmental TFs were positively associated with SCZ risk in our model (p = 1.64E-3, Fig 6C). A comparison between the enrichments of this GO term and generic GO terms is available as S6 Fig.

We further explored the developmental expression patterns of TFs that recognize SCZ-associated sequence motifs using single-cell RNA sequencing data from prenatal human cortex [39]. We found that many of the TFs that are highly associated with risk in our model are expressed in neuronal lineages, including members of the MEF2 family, the EMX family, the RAX family, and the DLX family (Fig 6D). Taken together, these results suggest a previously undescribed association between SCZ risk and AT-rich binding sites for neurodevelopmental TFs in enhancers of the fetal and adult brain.

## Discussion

Here, we developed tools and resources for Regulome-Wide Association Studies (RWAS), a flexible application of MAGMA for post-GWAS analyses of trait-associated enhancers and

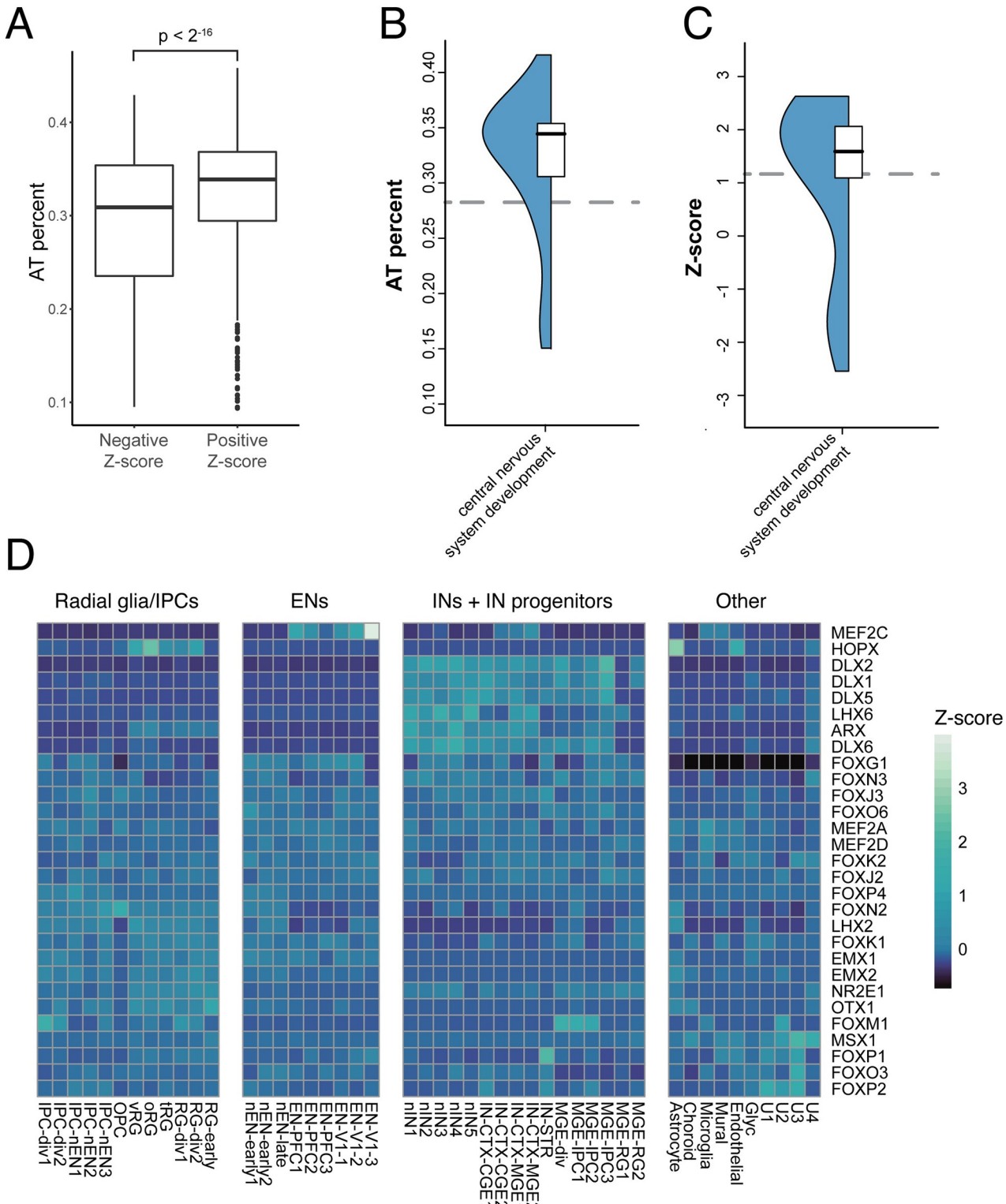

**Fig 6. TFs with AT rich motifs are overrepresented in risk-associated enhancers and have neurodevelopmental functions.** A) Positive association between the AT-richness of a given TF binding site motif and the effect size in the RWAS model. B) Higher median motif AT percentage of a given TF is positively associated with the TF being annotated to the Gene Ontology term "cell morphogenesis during neuron differentiation". C) TFs with higher median Z-score in the RWAS analysis are more likely to be annotated to "cell morphogenesis during neuron differentiation." Grey dashed line is the median value of the background set of all TFs in our dataset. D) Cell type-specific expression in the prenatal human brain for TFs that recognize positively associated motifs in the schizophrenia RWAS analysis. The displayed TFs recognize a motif with an RWAS Z-score > 3 in a brain enhancer. Each cell is colored by expression Z-scores averaged across the specified cell type.

**Table 1. Top 5 strongest positive associations between TF motif networks and schizophrenia risk in brain enhancers.**

| TF Motif | Z | Adj. P Value |
| --- | --- | --- |
| MEF2C-MA0497.1 | 3.54 | 4.05E-03 |
| ESX1-MA0644.1 | 3.48 | 5.08E-03 |
| LBX2-MA0699.1 | 3.46 | 5.40E-03 |
| MEF2A-MA0052.3 | 3.46 | 5.45E-03 |
| RAX-MA0718.1 | 3.40 | 6.79E-03 |

enhancer properties. Using these tools, we characterized enhancers associated with risk for schizophrenia.

Our analysis revealed a novel association of SCZ with AT-rich enhancers that are active in the human brain, many of which contain AT-rich sequence motifs recognized by neurodevelopmental TFs. Functional differences between AT-rich vs. GC-rich enhancers are not well understood. One previous study using Cap Analysis of Gene Expression showed that many enhancers actively transcribed in neurons are AT-rich and noted differences in TF occupancy in GC-rich vs AT-rich enhancers [40]. Our analysis generalizes this observation to a broader set of enhancers defined by independent epigenomic techniques. Functional differences of AT-rich vs. GC-rich promoters are better characterized, with AT-rich promoters containing distinct core promoter elements and serving different functions. For example, Lecellier et al. demonstrated that AT-rich promoter regions were disproportionately found near genes involved in the immune response [41]. Large genomic regions of relatively consistent nucleotide composition in the genome, known as isochores, have also been described to contain genes with shared functions; for example, GC-rich isochores tend to contain housekeeping genes, while AT-rich isochores tend to contain more tissue-specific genes [42]. To our knowledge, we are the first to report that neurodevelopmental TFs predominantly recognize AT-rich sequence motifs.

The specific neurodevelopmental TFs whose putative binding sites were enriched at SCZ-associated enhancers represent promising leads toward mapping the causal gene regulatory perturbations underlying SCZ. The most significant positive association in our TF RWAS was MEF2C-MA0497.1. This association is consistent with previous reports that the MEF2C motif is enriched at SCZ risk loci, and MEF2C target genes in the brain are enriched both for SCZ risk genes [43,44] and for genes differentially expressed in postmortem brain tissue from SCZ cases vs. controls. *MEF2C* itself is a positional candidate at an SCZ risk locus [5]. MEF2C is highly expressed in developing cortical excitatory neurons and is essential both for cortical neurogenesis and the modulation of cortical neuronal activity. Haploinsufficiency of MEF2C is known to cause a syndrome characterized by intellectual disability and neurological abnormalities [45]. Another network of interest is LBX2-MA0699.1, a motif recognized by multiple homeobox TFs. Of particular interest are EMX1 and EMX2, which are highly expressed in the developing dorsal telencephalon in the lineage leading to excitatory neurons and have well-established roles in cortical thickness and arealization [46–49]. Similarly to MEF2C, the area containing the gene for EMX1 is itself a candidate schizophrenia risk locus [5]. Mutations in EMX2 have been noted in patients with severe schizencephaly [50]. The LBX2-MA0699.1 motif is also recognized by members of the DLX and ARX families. Unlike the EMX factors that are involved in excitatory neuron development, these TFs are critical for inhibitory neuron development and migration [51, 52]. Mutations in ARX have been linked to cases of X-linked lissencephaly with abnormal genitalia in humans [53]. A limitation of our analysis is

that motif-based predictions cannot resolve the specific members of this TF family that occupy the SCZ-associated enhancers, but the family as a whole merits increased attention in SCZ.

Understanding the gene regulatory mechanisms underlying risk for polygenic traits is a complex task. Our RWAS framework is complementary to existing tools and is uniquely suited to test associations between the characteristics of specific regulatory elements and disease risk. While our current approach is focused on testing associations of enhancers with common SNPs identified through GWAS, annotating gene regulatory consequences of rare non-coding single-nucleotide variants and copy-number variants represents an important future direction [54].

RWAS is readily applicable to additional traits of interest, as it is implemented with a widely used software tool (MAGMA) and requires only GWAS summary statistics and enhancer level annotations. We have made our instructions for running RWAS available at www.github.com/casalex/RWAS. We have also made available the enhancer annotations for all 127 ROADMAP samples, and similar enhancer models suitable for RWAS are now available from >800 samples from the ENCODE consortium, spanning all of the major human organs and tissues [55].

## Methods

### Enhancer download and processing

Predicted enhancer regions were derived from 25-state ChromHMM [16] chromatin state models downloaded from the ROADMAP consortium website (https://egg2.wustl.edu/roadmap/web_portal/imputed.html). We defined enhancers by pooling nine states from these models: 1) transcribed 5' preferential and enhancer, 2) transcribed 3' preferential and enhancer, 3) transcribed and weak enhancer, 4) active enhancer 1, 5) active enhancer 2, 6) active enhancer flank, 7) weak enhancer 1, 8) weak enhancer 2, and 9) primary H3K27ac possible enhancer. Enhancer annotations from the psychENCODE consortium were downloaded from http://resource.psychencode.org/. The brain enhancers used in the schizophrenia RWAS analyses were E067 (Brain Angular Gyrus), E068 (Brain Anterior Caudate), E069 (Brain Cingulate Gyrus), E070 (Brain Germinal Matrix), E071 (Brain Hippocampus Middle), E072 (Brain Inferior Temporal Lobe), E073 (Brain_Dorsolateral_Prefrontal_Cortex), E074 (Brain Substantia Nigra), E081 (Fetal Brain Male), and E082 (Fetal Brain Female).

Enhancer annotations used for partitioned heritability and RWAS analyses were pre-processed in a uniform pipeline. Enhancer boundaries are often poorly defined, and MAGMA and similar tools suffer from length bias wherein long regions with many SNPs have anti-conservative p-values (S7 Fig). To overcome these issues, our analyses were conducted using 1 kb enhancer centroids. Enhancer regions were merged with any directly adjacent annotations, and the center of each merged region was determined. The boundaries were then extended by 500 bp upstream and downstream of this center, resulting in a 1kb region centered on the middle of the enhancer region. Any enhancers falling within the MHC region or ENCODE blacklist regions [56] were removed.

### Jaccard similarity

In order to compare enhancer similarity across all 127 samples we computed pairwise genome-wide Jaccard distances using the BEDtools software suite [57]. Groupings were determined using hierarchical clustering.

### GWAS summary statistics

We retrieved GWAS summary statistics for schizophrenia [5] from the Psychiatric Genomics Consortium data portal (https://www.med.unc.edu/pgc).

## Partitioned heritability

Stratified LD score regression (LDSC version 1.0.1) was applied to GWAS summary statistics to evaluate the enrichment of trait heritability across the 127 enhancer sets [3]. These associations were adjusted for 52 annotations from version 1.2 of the LDSC baseline model (including genic regions, enhancer regions and conserved regions).

## Observed versus expected overlaps

We performed three different overlap analyses to determine observed vs expected overlaps of enhancer annotations. We first took enhancer annotations and shuffled their positions, taking care to exclude the MHC region and any ENCODE blacklist regions [56]. We then obtained MPRA SNP locations [19] and ENCODE ChIP-seq peaks from human brain middle frontal area 46 (H3K27ac, ENCSR554HDT; H3K9me3, ENCSR349III) [58]. These annotations were overlapped with shuffled and non-shuffled enhancers to obtain expected and observed overlap counts for each annotation.

## RWAS

RWAS was performed using the linear model implemented in MAGMA's covariate mode. This was accomplished by using the enhancer sets in place of genes. The processed enhancers were supplied as a genomic location file format as described in the MAGMA manual, and enhancer-level attributes were supplied as continuous covariates. GERP hg19 phylogeny scores were downloaded from http://hgdownload.cse.ucsc.edu/goldenpath/hg19/phastCons100way/ and averaged across each enhancer region to yield a conservation score for each enhancer for association testing. TSS for each gene were taken from a supplied MAGMA gene file (https://ctg.cncr.nl/software/magma). Distance to the nearest TSS for each enhancer was determined using the BEDTools "closest" command, and this distance was supplied to MAGMA as a covariate for association testing. HAR regions were downloaded from Supplementary Table 1 of Doan et al. (2016) [59]. These regions were expanded by 2,500 bp upstream and downstream before being intersected with the enhancer regions, yielding a binary measure for each enhancer indicating if an enhancer overlapped an HAR or not. This was input as a covariate in the MAGMA analysis. Similarly, HGEs were defined as differentially enriched CREs between human and rhesus macaque from Vermunt et al. and overlaps were tested for association [60].

Chromosomal contact testing was performed using the set analysis in MAGMA. We used HiC from the cortical plate of the developing human brain [22] to assign genes to enhancers that they physically contact. Enhancers that contact genes with a given ontology term were assigned to the enhancer set for that term, and the resultant enhancer sets were tested for association with risk using MAGMA's gene set mode. The gene sets are available in S1 Table and were derived from the following datasets, as we have described previously [61]: genes intolerant of loss-of-function variants from gnomAD (pLI > = 0.9 or LOEUF deciles 1 or 2) [62]; risk genes from studies of rare variants in four disorders, including severe developmental disorder risk genes from the Deciphering Developmental Disorders consortium's DDG2P database (Disorders of Brain Development) [63], autism spectrum disorder risk genes from the Autism Sequencing Consortium (Autism risk [exomes]) [64], bipolar disorder risk genes from the BipEx Consortium [65]; genes identified from large-scale GWAS, identified by gene-based analyses with MAGMA [9] (p < 2.77e-6 unless noted as FDR, in which case adj. p < 0.05) for bipolar disorder [66], major depression [67], and neuroticism [68], differentially expressed genes in the prefrontal cortex of individuals with schizophrenia, bipolar disorder, and autism from the PsychENCODE consortium [69] (http://resource.psychencode.org/Datasets/Derived/DEXgenes_CoExp/DER-13_Disorder_DEX_Genes.csv); genes associated with schizophrenia

from SCHEMA [70]; target gene networks of the neuropsychiatric risk genes FMRP, RBFOX1/ 3, RBFOX2, CHD8, CELF4, and microRNA-137 derived from functional genomics experiments, annotated by Genovese et al. [71]; synaptic genes, including genes from SynaptomeDB; proteins localized to the axonal growth cone, and genes annotated to the Gene Ontology term "neuron spine" [72,73]. In-text p-values were derived by taking the minimum p-value across the 10 brain enhancer annotations and adjusting for the number of annotations.

## TF binding site RWAS

Brain-specific DNAse-seq footprints annotated with matching TF motifs were obtained from our previously described footprint atlas [37]. The HINT atlas was used due to its superior performance in TF binding site prediction. A HINT score cutoff of 55 was used to filter out low-quality footprints. We limited our analysis to the 503 JASPAR vertebrate core motifs that had mappings to human TFs. Footprints that fell within the boundaries of a given enhancer were annotated to that element, yielding a covariate file containing counts of each motif for each enhancer. A total binding site control was used to control for total binding site number. MAGMA was run in the covariate mode as described above. Meta-analysis of adult and fetal brain enhancer RWA analyses was performed by taking the highest absolute value Z-score from the individual enhancer RWA results for each motif. The resultant p-values were adjusted for the number of results meta-analyzed (10).

## Motif to TF mapping

The footprint-motif pairs were mapped to TFs using a key described in our previous work [37]. These mappings were restricted to JASPAR motifs, so only these motifs were included in downstream analyses.

## GO term analysis

We used the Wilcoxon rank-sum test as implemented in the R package GOfuncR to test for association between TF function and scores/attributes from our models.

## Single-cell RNA-seq analysis

The single-cell RNA-seq dataset from the prenatal human cortex was downloaded from the UCSC cell browser (http://cells.ucsc.edu/cortex-dev/exprMatrix.tsv.gz) [38]. Any TF with an RWAS Z-score that was expressed in $> 250$ cells in this dataset was included in the analysis. The expression Z-scores were generated using R's scale() function, grouped by cell type, then averaged.

## Supporting information

**S1 Fig.** Observed vs expected epigenetic mark ChIP-seq peak overlaps of 10 chromHMM brain enhancers A) Enhancer marker H3K27ac peaks from middle frontal area 46 are enriched in chromHMM brain enhancers B) Heterochromatin marker H3K9me3 peaks from middle frontal area 46 are depleted in chromHMM brain enhancers.
(TIFF)

**S2 Fig. Enhancer annotation summary statistics.** A) Genomic coverage by tissue category. B) Adult brain and astrocyte enhancer annotations had the highest genomic coverage compared to fetal, neurosphere, and ESC-derived enhancer annotations. C) Enhancer number by tissue category. D) Adult brain and astrocyte enhancer annotations had the highest enhancer number compared to fetal, neurosphere, and ESC-derived enhancer annotations. E) Enhancer

length by tissue category. F) Fetal brain and neurosphere enhancer annotations had the highest mean enhancer length compared to adult brain, astrocyte, and ESC-derived enhancer annotations. G) Enhancer number and percent genomic coverage are tightly associated (p = 8.1E-59). H) Enhancer length and genomic coverage are not associated (p = 0.64). I) Enhancer length and enhancer number are negatively correlated (p = 9.1E-6).
(TIF)

**S3 Fig. Jaccard similarity between all 127 chromHMM enhancer annotations.** Annotations are arranged by hierarchical clustering. Differential enhancer utilization between tissues clusters samples by organ. Brain enhancers largely cluster together, with the exception of ESC-derived cells and astrocytes. Interestingly, while the female fetal brain and male fetal brain were in the same cluster, the female fetal brain enhancers were slightly more similar to the neurosphere samples than to the male fetal brain samples. This is likely due to technical differences, as fetal male brain (E081) is the only sample from this cluster where the primary tissue was from the Broad Institute, while all other neurosphere/fetal samples were from UCSF. The differences in sample origin did not have a major effect on the overall cluster structure, as E081 did not cluster with any of the other Broad Institute samples such as H9 derived neuronal progenitor cultured cells (E009), H9 derived neuron cultured cells (E010), or NH-A Astrocytes (E125).
(TIF)

**S4 Fig. Enhancer annotations vary by length and nucleotide composition.** A) Fetal brain and neurosphere enhancer annotations are underrepresented in the lowest length bins compared to other brain samples. B) Fetal brain and neurosphere enhancer annotations have more super-long enhancers than other brain enhancer annotations. C) AT percentage of enhancer annotations by tissue. D) Adult brain enhancer annotations have slightly higher AT richness compared to fetal brain and neurosphere enhancer annotations. E-F) On average, super-enhancers in the brain (>10 kb, 'TRUE') tend to be more AT rich than enhancers of shorter length ("FALSE").
(TIF)

**S5 Fig. Association between AT-richness and schizophrenia risk across all ChromHMM enhancers aggregated by tissue.** Brain enhancers had the strongest association, with germinal matrix (E070) having the most associated individual annotation.
(TIF)

**S6 Fig. Reproduction of [Fig 6] with null hypothesis GO terms for comparison.** A) Higher median motif AT percentage of a given TF is positively associated with the TF being annotated to the Gene Ontology term "cell morphogenesis during neuron differentiation" but not general GO terms (biological processes, cellular components, molecular function) B) TFs with higher median Z-score in the RWAS analysis are more likely to be annotated to "cell morphogenesis during neuron differentiation" and are not more likely to be annotated to general GO terms.
(TIF)

**S7 Fig. Non-truncated enhancers suffer from length bias in MAGMA gene-set analyses.** A. Z-scores are higher in longer enhancers compared to shorter enhancers in the PGC2 schizophrenia GWAS. B. Z-scores show similar inflation in long enhancers in 75 unrelated UK Biobank traits.
(TIF)

**S1 Table. Gene lists for schizophrenia RWAS gene set testing.**
(TSV)

**S2 Table. Meta-analyzed and single enhancer annotation TF schizophrenia RWAS results in the adult and fetal brain.**
(XLSX)

**S3 Table. Genome-wide significant enhancer hits within SCZ risk loci across 127 human tissues.** Sample names are as described in the ROADMAP epigenomics project integrative analysis portal: https://egg2.wustl.edu/roadmap/web_portal/meta.html.
(ZIP)

**S4 Table. Untransformed LDSC result p-values.**
(XLSX)

# Acknowledgments

We thank the University of Maryland Medical Scientist Training Program for their ongoing assistance and mentorship.

# Author Contributions

**Conceptualization:** Alex M. Casella, Seth A. Ament.

**Data curation:** Alex M. Casella.

**Formal analysis:** Alex M. Casella.

**Funding acquisition:** Seth A. Ament.

**Investigation:** Alex M. Casella, Seth A. Ament.

**Methodology:** Alex M. Casella, Carlo Colantuoni, Seth A. Ament.

**Project administration:** Seth A. Ament.

**Resources:** Seth A. Ament.

**Software:** Alex M. Casella.

**Supervision:** Carlo Colantuoni, Seth A. Ament.

**Visualization:** Alex M. Casella.

**Writing – original draft:** Alex M. Casella, Seth A. Ament.

**Writing – review & editing:** Alex M. Casella, Carlo Colantuoni, Seth A. Ament.

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
