## [Decision Letter · Decision Letter 0]

18 Oct 2021

Dear Assistant Professor Ament,

Thank you very much for submitting your manuscript "RWAS: Identify enhancer properties associated with genetic risk for complex traits" for consideration at PLOS Computational Biology.

As with all papers reviewed by the journal, your manuscript was reviewed by members of the editorial board and by several independent reviewers. In light of the reviews (below this email), we would like to invite the resubmission of a significantly-revised version that takes into account the reviewers' comments.

We cannot make any decision about publication until we have seen the revised manuscript and your response to the reviewers' comments. Your revised manuscript is also likely to be sent to reviewers for further evaluation.

Sincerely,

Teresa M. Przytycka

Associate Editor

PLOS Computational Biology

Sushmita Roy

Deputy Editor

PLOS Computational Biology

Reviewer's Responses to Questions

**Comments to the Authors:**

Reviewer #1: This is a high quality study, well-written, with very interesting findings!

Three comments:

1. The authors used a highly comprehensive approach to define enhancers, particularly for the brain. But almost all data used were from ChIP-seq. I understand that those are good indicators of regulators overall, but the precision at individual genes might still be questionable. Given gene expression is the ultimate product of the regulation, why gene expression information was not considered.

I would love to see one additional analysis to show how well the expression activity is related to the states of the predicted regulators. It might be a challenge to combine regulators across a long genomic region to predict expression. I'd like to know how the authors address this issue or their thoughts about this problem.

In the end, I really hope to see that the enhancers are meaningful in relation to gene expression instead of using one histone pattern to prove another. I am not completely convinced about the current way defining enhancers.

2. The authors used "enhancer" for the functional elements they defined in the paper. Are they really all enhancers? Please clarify. Would the other types of regulators be relevant too? or they just used enhancers to represent all regulators, including silencers and insulators.

3. On page 14, the authors used MPRA and eQTL data to support a few top RWAS signals. I think it is more important to have an overall evaluation, like an overlap or enrichment test, to show how many or % of RWAS signals can be validated. eQTL is particularly interesting.

Again, I just want to congratulate the authors for the wonderful work. Hope it can be published soon.

Reviewer #2: In their paper “RWAS: Identify enhancer properties associated with genetic risk for complex traits,” Casella et. al present an alternative usage of MAGMA for scoring enhancers for enrichment in GWAS studies. While the general idea of something like RWAS isn’t completely novel, and the paper doesn’t feature any major methods development, the application here is well done and thorough. I have the following, mostly minor, items the authors should address:

1. I find the title a little confusing. It would be easier to read if it specified that RWAS is a method for identifying those properties. Additionally, the paper seems much more focused on downstream findings than the RWAS method itself.

2. Figures S1 E&F should say average enhancer length.

3. Is there a good reason why fetal female and fetal male don’t cluster together (Figure 2)? Would be good to discuss this in the text.

4. Looking at Figure 3b, there appear to be brains with nearly identical levels of significance which is surprising. Is there a good reason for this?

5. Figures 4a and b are informative, but they are not well described in the text (also I think the first mention of 4b should be 4a bottom).

6. It is claimed that “Examination of specific risk loci indicated that risk-associated enhancers capture the genetic risk signal at many of the SNP-based risk loci in a tissue-specific manner” [line 197] but In this example I also see significant schizophrenia risk in enhancers in primary t-cells and not temporal lobe. Is this enhancer also a true positive? Furthermore, it would be good to have some summary statistics of hits in enhancers across tissues (e.g. fraction of genome-wide significant hits captured by each enhancer set).

7. The MPRA analysis is quite ad-hoc. It needs numbers for how many overlapping hits were observed and how many would be expected.

8. In Figure 5a, do stars represent significance? Also, how was this subset of the 64 gene sets selected? It would be good for some gene sets that would be expected to be non-significant to be shown for comparison as negative controls.

9. Figures 6a and 6b, should show some other GO terms for comparison. A negative control would also be good.

10. In Figure 6d, is this the full set of TFs that recognize positively associated motifs? If not, how were they selected? Also, it's hard to tell by looking at the figure if expression levels are significant.

11. The GitHub for RWAS is very limited, I would like to see code for some downstream analyses.

Reviewer #3: In this paper by Casella et al. author propose an approach denoted RWAS for predicting enhancer associated with complex traits and disorders. The proposed approach uses the tool MAGMA that was originally built for gene based GWAS study. Predicting the enhancer associated and contributing to each complex disorder is an important problem. I do have several major comments:

1. There seems to be no new code/method developed here (https://github.com/casalex/RWAS), simply commands to run MAGMA for this application. Please rewrite the paper to make it clear this is not a novel computational method but just a new application of MAGMA.

2. The proposed approach is based on running MAGMA. However, MAGMA is developed for studying genes. Utilizing same approach for application to enhancers might have some unforeseeable complications. For example, the fact that multiple enhancers can have complicated non-linear relationships with gene expression of multiple genes. How would this impact the result of the method?

3. There are several methods developed that also calculated the association of enhancer with diseases (e.g., FENRIR or ABC model). Please do a comparison with other computational methods.

4. There are well known CNVs correlated with schizophrenia. Is there an enrichment of these CNVs impacting the predicted enhancer?

5. There are studies that have shown significant enrichment of denovo coding variants in SCZ cases (Gulsuner et al. 2013). Can the author find some sequencing data that shows significant enrichment of rare variants in affected cases impacting these enhancers?

**Have the authors made all data and (if applicable) computational code underlying the findings in their manuscript fully available?**

Reviewer #1: Yes

Reviewer #2: None

Reviewer #3: Yes

PLOS authors have the option to publish the peer review history of their article (what does this mean?). If published, this will include your full peer review and any attached files.

Reviewer #1: **Yes: **Chunyu Liu

Reviewer #2: No

Reviewer #3: No
---

## [Decision Letter · Decision Letter 1]

7 Jun 2022

Dear Assistant Professor Ament,

Thank you very much for submitting your manuscript "Identifying enhancer properties associated with genetic risk for complex traits using regulome-wide association studies" for consideration at PLOS Computational Biology. As with all papers reviewed by the journal, your manuscript was reviewed by members of the editorial board and by several independent reviewers. The reviewers appreciated the attention to an important topic. Based on the reviews, we are likely to accept this manuscript for publication, providing that you modify the manuscript according to the review recommendations.

Sincerely,

Teresa M. Przytycka

Associate Editor

PLOS Computational Biology

Sushmita Roy

Deputy Editor

PLOS Computational Biology

[LINK]

Reviewer's Responses to Questions

**Comments to the Authors:**

Reviewer #1: All my questions have been well addressed.

Reviewer #2: The authors have addressed all major points raised in the initial review and added several interesting analyses based on reviewer comments. I only have a few minor comments:

- The authors raise an interesting point about the fetal female brain sample. Given the technical differences, I wonder if it may be worth excluding that sample. I leave it up to the authors.

- line 222 - “male fetal brain” is missing the word “enhancers”:

- Regarding figure 5a, I still don't see how a subset of the 64 were selected.

Reviewer #3: The authors have adequately addressed my comments.

**Have the authors made all data and (if applicable) computational code underlying the findings in their manuscript fully available?**

Reviewer #1: Yes

Reviewer #2: Yes

Reviewer #3: Yes

PLOS authors have the option to publish the peer review history of their article (what does this mean?). If published, this will include your full peer review and any attached files.

Reviewer #1: **Yes: **Chunyu Liu

Reviewer #2: No

Reviewer #3: No

Figure Files:

Data Requirements:

Reproducibility:

References:

---

## [Editor Report · Decision Letter 2]

23 Jul 2022

Dear Assistant Professor Ament,

We are pleased to inform you that your manuscript 'Identifying enhancer properties associated with genetic risk for complex traits using regulome-wide association studies' has been provisionally accepted for publication in PLOS Computational Biology.

Best regards,

Teresa M. Przytycka

Associate Editor

PLOS Computational Biology

Sushmita Roy

Deputy Editor

PLOS Computational Biology

---

## [Editor Report · Acceptance letter]

2 Sep 2022

PCOMPBIOL-D-21-01269R2 

Identifying enhancer properties associated with genetic risk for complex traits using regulome-wide association studies

Dear Dr Ament,

I am pleased to inform you that your manuscript has been formally accepted for publication in PLOS Computational Biology. Your manuscript is now with our production department and you will be notified of the publication date in due course.

With kind regards,

Anita Estes
